# Emerging Technologies and Their Link to Digital Competence in Teaching

**Miguel Ángel García-Delgado** [1,*] , **Sonia Rodríguez-Cano** [1] , **Vanesa Delgado-Benito** [1]
**and María Lozano-Álvarez** [2]

[1] Faculty of Education, Department of Education, University of Burgos, 09001 Burgos, Spain;
srcano@ubu.es (S.R.-C.); vdelgado@ubu.es (V.D.-B.)
[2] Faculty of Social and Legal Sciences, Catholic University of Ávila, 05005 Ávila, Spain; maria.lozano@ucavila.es
[*] Correspondence: mgd0111@alu.ubu.es

**Abstract:** The new educational reality requires teachers to have a series of skills and competences that allow them to improve the teaching–learning process and therefore the quality of teaching, integrating technology and emerging technologies. In order to assess the competence level of teachers, a descriptive study was designed, in which 101 teachers from different stages and geographical locations in Spain took part and were administered the DigCompEdu Check-in questionnaire. The results show average levels of teachers' digital competence (B1 and B2, particularly), and an incipient use of emerging technologies by teachers, with less than 50% of the sample not using these technologies in their daily classroom activities, although those who show a higher level of digital competence are also those who integrate them more in their daily work. The results correspond with similar studies, corroborating the average level of teachers' digital skills.

**Keywords:** teachers; virtual reality; augmented reality; mixed reality; digital competence

## 1. Introduction

New technologies have brought about a remarkable change in our daily lives, from the way we interact, communicate, or enjoy our leisure time to the way we obtain information and knowledge, making them much more accessible [1]. Technologies have gained importance in our daily lives [2], considering the special relevance of the modifications they have made to our world, which leads to the need to train people to make them digitally competent in order to support their engagement with society and the current reality. In this vein, we can highlight the importance of digital competence in meeting the challenges posed by our knowledge-based society since, thanks to this competence, the skills and abilities possessed by individuals can be revealed [3].

At present, a classification of people can be established based on their adaptation to the use of ICTs, thus creating two clearly differentiated groups [4]: digital natives, those who were born with the new technologies already consolidated at a social level, who can be further differentiated between those who were born before the existence of the Internet as we know it today, and those who were born with this system already fully established; and digital immigrants, those who had to adapt and develop their skills at the time when the technological revolution took place.

The importance of new technologies has become evident in the field of education, which has led to the emergence of different positions among teachers in relation to new technologies, establishing a clear distinction between those who adopt technology, including it in their teaching style in order to improve the teaching–learning process, and those that are reluctant to introduce it in the classroom. However, the main problem is not only in the type of teacher, but in the need to transform the teaching practice in order to adapt it to the new reality brought about by the emergence of new technologies [5].

Virtual reality (VR) and augmented reality (AR) have substantially modified education in the 21st century [6], although their boom has taken place in the last ten years, conquering different milestones such as the accessibility of this type of technology to the general public. All this change was accelerated by the health crisis that occurred in 2020, which precipitated teachers to immerse themselves in a more intensive use of the tools made available by technological advances. Based on this study [6], nearly 95% of teachers consider that the health crisis was a turning point in the use of technologies to carry out their teaching work, and they also consider that they have increased their skills related to the creation of digital content due to the urgent renovation that they had to carry out at that time to ensure that their teaching was adapted to the moment they were living through. Furthermore, one of the main objectives of the European Union's 2030 Agenda is to promote and democratise accessibility to digital tools and content.

It is also important to highlight the importance of media literacy as an important element in the digital competence training of teachers in the context of the reality brought about by the inclusion of technologies in the classroom, a reality that is reflected both in educational policies and in the scientific literature on the subject [7]. In this vein, it should be highlighted that teachers are currently required to develop new competences in order to maximise their work in the classroom; therefore, it is assumed that they are capable of incorporating technologies as part of their strategies to improve and favour the teaching–learning processes and their own teaching work [8].

Therefore, the term Digital Competence in Teaching (DCT) has emerged, a concept that encompasses a broader reality than that which refers to training processes and the way in which technologies are used in the classroom [9]. Digital Competence in Teaching [1] does not only refer to the use of technology in the classroom, its use to optimise the possibilities it offers, and the teaching–learning process; it also refers to the environment in which the different learning situations and experiences take place. The need for training that students require is also noteworthy. Therefore, teachers should provide them with the necessary resources and tools to enable them to actively participate in the digital society and to be ready to work in this new reality [1]. Additionally, the legislation in force—through the open curriculum—allows learning not to be circumscribed exclusively in the classroom. Quite the opposite, it is open to the possibility of it taking place in different environments, a space in which new technologies are particularly relevant [10].

In short, both teachers and educational policies tend to set digital competence for teachers as one of the most frequent goals, and despite the progress that has been made, these goals have never been fully achieved [11]. Moreover, in recent years there has been a remarkable growth and interest in the field of educational research on digital competence in teaching [12], both at a national and an international level, which will favour the creation of elements that will encourage the achievement of goals in this respect in teacher training.

The key role of Information and Communication Technologies in teaching is undeniable, but progress in these technologies has meant that teaching–learning spaces have been substantially modified, and it is here where the so-called emerging technologies, including Augmented Reality, Virtual Reality, and Mixed Reality, have taken on special relevance, and begun to position themselves as important tools in the classroom [13], with the aim of becoming a means to develop the methodology, but also of promoting different learning situations.

In view of the above and taking as a basis the Framework of Reference for Digital Competence in Teaching [14], relating it to the European Commission's Digital Competence Framework for Teachers (DigCompEdu) [1], this study attempts to assess the level of digital competence of teachers at different educational stages and show an approximate reality of the current state of Digital Competence in Teaching, relating it to the use of emerging technologies.

For this reason, some of the most relevant related studies carried out by Vera and García-Martínez [5], Cabero-Almenara and Palacios-Rodríguez [7], Torres-Barzabal et al. [8],

Cabero-Almenara et al. [9], Cristóvão et al. [10], and Casal-Otero et al. [11] were used as references.

## 2. Materials and Methods

This research is descriptive in nature and its main objective is to assess the impact of emerging technologies on teachers' digital competence in teaching. Secondary objectives are also pursued, such as assessing teachers' level of digital competence and finding out what kind of emerging technologies they use and to what extent they are able to use them.

In order to respond to the objectives set and collect data, the "DigCompEdu Check-In" questionnaire, a tool that fosters self-evaluation and reflection of teachers on aspects related to new technologies, was telematically implemented; specifically, the version translated into Spanish [7]. The tool used is made up of twenty-two items that are subdivided into six fields consisting of the following:

- Professional engagement: relating to organisational communication, professional collaboration, reflective practice, and teachers' digital literacy.
- Digital assets: the ability to select, create, and modify assets and to manage, exchange, and protect data.
- Digital pedagogy: how, when, and why to use digital technologies to help maximise their benefits, as well as the ability to monitor activities in collaborative environments and foster collaborative and self-directed learning.
- Assessment and feedback: skills relating to assessment strategies, the analysis of evidence, and evidence for the identification of learners in need of additional support, as well as the use of technology to provide feedback to learners.
- Empowering learners: favouring accessibility and inclusion of learners with regard to new technologies, offering a variety of options to encourage differentiation and personalisation of learning according to the qualities of the learners, and the use of new technologies in order to increase the active participation of learners in the classroom.
- Facilitating students' digital competence: establishing the teacher's ability to teach students how to assess the trustworthiness of information searched online, to foster digital communication, collaboration, and the creation of digital content through responsible and safe online use, and to encourage creative problem solving using new technologies.

In order to measure and evaluate the items in the different fields, a Likert scale of five intervals was used, which allows teachers to reflect on their educational practice by selecting the extent to which they identify with the given statement [7]. In addition to the dimensions described above, the online questionnaire collected socio-demographic data: gender, age, teaching experience, type of contract that links them to their educational institution (permanent, temporary, associate or full-time), type of school where they work, perceived socio-economic level of students, whether their school participates in digitisation programmes, the hours they dedicate to the use of technology in the classroom, the digital tools used, the perceived level of digital citizenship competence, whether they actively use social networks, and the working conditions that favour the use of new technologies. A new section on emerging technologies was also added, asking whether they have used any type of emerging technology (augmented, virtual, or mixed reality), and the extent to which they have used it.

In order to calculate the level of Digital Competence in Teaching, the levels set were taken as a reference [7], assessing the answers given from 0 to 4 points. This allows a level to be established for each of the fields as well as one at a global level; these are grouped into three categories, low level: A1 (beginner) and A2 (explorer); intermediate level: B1 (integrative) and B2 (expert); and high: C1 (leader) and C2 (pioneer).

In order to collect data properly, the validated tool was distributed in electronic format, requesting participation on a voluntary basis, and guaranteeing at all times the data protection of the participants and their responses, ensuring their anonymity.

The sample for this study comprised a total of 101 teachers from different locations, namely Spain, Portugal, and Chile (Table 1), of whom 31.68% were men and 66.34% were women, while 1.98% preferred not to say.

**Table 1.** Frequency analysis by gender and nationality.

| Nationality | Men | | Women | | I Would Rather Not Answer | | Total | |
|---|---|---|---|---|---|---|---|---|
| | N | % | N | % | N | % | N | % |
| Spain | 26 | 25.74% | 63 | 62.37% | 2 | 1.98% | 91 | 90.09% |
| Portugal | 6 | 5.94% | 0 | 0% | 0 | 0% | 6 | 5.94% |
| Chile | 0 | 0% | 4 | 3.97% | 0 | 0% | 4 | 3.97% |
| Total | 32 | 31.68% | 67 | 66.34% | 2 | 1.98% | 101 | 100% |

## 3. Results

As shown in Figure 1, the results of the overall level of competence of the participants in the sample in Digital Competence are shown. Overall, 80% of the participating teachers had an intermediate, upper-intermediate, or high level, distributed as follows: the majority of the participating teachers had an intermediate level, identifying themselves as an integrator (B1), in which 38% of participants are included; just over 25% of the teachers were identified as an upper-intermediate level, as experts (B2); only 13% of the sample considered themselves as leaders (C1); and if we refer to the pioneers (C2), only 4% of the sample is represented at this level. In contrast, regarding the lowest levels, 2% of the sample considered themselves as beginners (A1), and less than 16% considered that they would be classified as explorers (A2). In general terms, we can see that the teachers who took part in this research fall into the medium and high levels, with the lowest level being far behind, with hardly any representation, and the highest level, which implies total mastery by the teachers in the fields investigated, also poorly represented. To sum up, we can say that in general, the participating teachers had good competence levels in Digital Competence in Teaching.

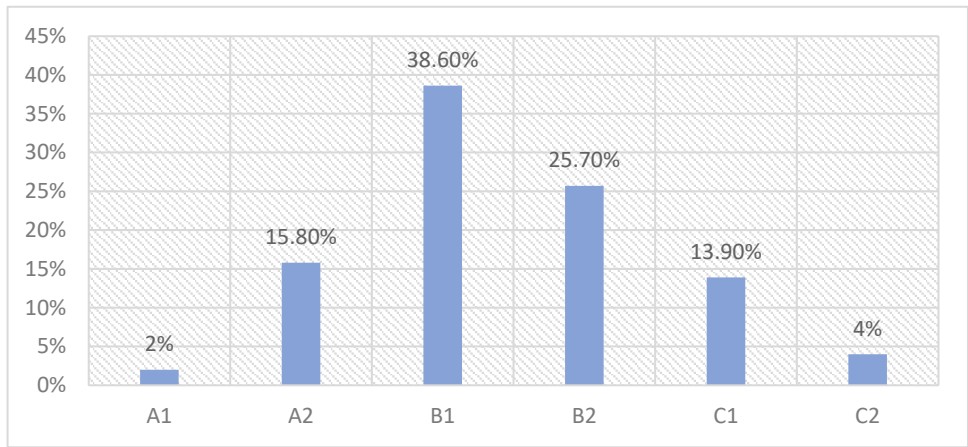

**Figure 1.** Overall ranking in Digital Competence in Teaching.

Based on the averages obtained (Table 2), it can be seen that the fields of competence in which the teachers participating in the study stand out the most are: use of digital resources (3.29), professional commitment (3.14), student empowerment (3.12), and digital pedagogy (3.01). The weakest fields of competence are assessment and feedback (2.94) and facilitating students' digital competence (2.63), the latter being the lowest average. Therefore, it is clear that the complexity of the use of technologies does not lie in their use, but rather that the main weaknesses in their use lie in carrying out assessment processes, relegating technologies to the background in favour of more traditional methods and tools. Likewise,

the need to use technologies for the development of students' digital competence in a more practical and useful way, leaving aside the theory in this regard, is evident.

**Table 2.** Values obtained in each field of competence.

| Field of Competence | Average | Standard Deviation |
|---|---|---|
| Professional Commitment | 3.14 | 1.087 |
| Digital Resources | 3.29 | 1.186 |
| Digital Pedagogy | 3.01 | 1.330 |
| Assessment and Feedback | 2.94 | 1.318 |
| Empowering Students | 3.12 | 1.525 |
| Facilitating Students' Digital Competence | 2.63 | 1.231 |

For this reason, the results will be detailed by categories; firstly, those relating to digital competence in teaching according to gender, age, the educational stage in which they work, teaching experience, and type of educational centre which they carry out their teaching activity. Subsequently, the relationship between the level of digital competence and the use of emerging technologies, as well as the skill level in the use of these technologies, will be established.

### 3.1. Results by Gender

In terms of gender, as we can see in Figure 2, those who preferred not to indicate their gender obtained an average score and identified with the expert level (B2). Likewise, there are notable differences between women and men, especially at levels A2, B1, and B2, with women obtaining higher scores than men at the three levels specified above. On the other hand, it should be noted that results are even at the lowest level (A1) and at the two highest levels (C1 and C2). Overall, it can be seen that the majority of the sample, around 78% have a medium, medium-high, or high level of digital competence (levels between B1 and C1), which indicates that most of them have sufficient skills in this respect to be able to implement the use of technologies in their teaching activity.

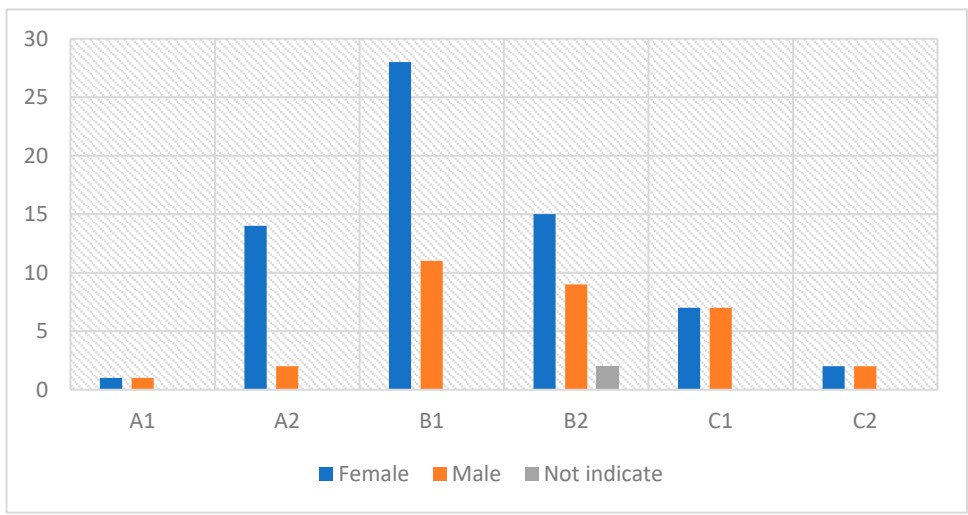

**Figure 2.** Level of Digital Competence by Gender.

### 3.2. Results by Age

In the data referring to age, we found that the majority of the sample is between B1 and C1 levels, with both extremes, low and high proficiency, being relegated to a residual role (Figure 3).

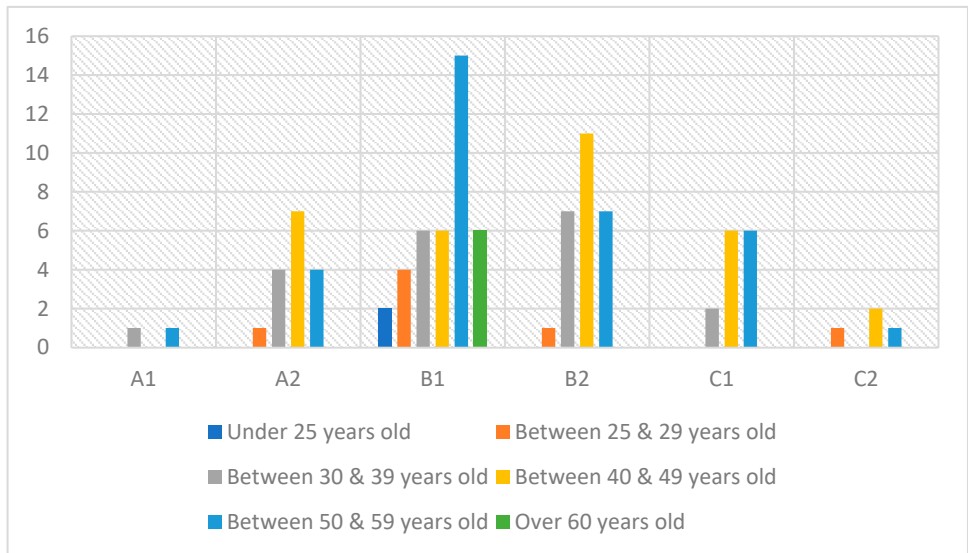

**Figure 3.** Level of Digital Competence by Age.

In terms of age ranges, all participants under the age of 25 are represented in the B1 level. On the other hand, if we look at those between 25 and 29 years of age, their presence at level C2 is striking, as is their absence at level C1, immediately above, and A1, the most basic level; although we note that despite these, the participants in this age range have a medium level of digital competence in teaching (B1 and B2).

Referring to the population between 30 and 39 years of age, we found that they are one of the groups at the lowest level (A1), and they are not represented at the highest level (C2); despite this, they show medium-high levels of competence, with a wide representation at B1 and B2 levels, as well as at A2 level and somewhat lower at C1 level. Participants aged between 40 and 49 are most represented at levels A2, B2 and C2, as well as at level C1, as are those in the 50–59 age range; their absence at level A1 is noteworthy in this group, and we observe that they show an upper-intermediate and high level of proficiency, respectively.

If we refer to those teachers in the age range between 50 and 59, they are one of the groups with representation at levels A1 and C2; they have a large representation at level B1, being the most represented group at this level. Finally, for participants over 60 years of age, all of them are located at the lowest level of the intermediate position (B1).

To sum up, we could not establish a significant difference between the different ages, as it is not a key factor in the level of Digital Competence of Teaching.

### 3.3. Results by Educational Stage

With regard to the existing level of competence at the different educational stages (Figure 4), we will proceed to describe it according to the level at which they teach. As far as early childhood education teachers are concerned, most of them have an intermediate level (B1), with a minority at level (B2) and without reaching the highest values; a fact which is notably different from the rest of the teachers who at least in some cases manage to reach the highest levels and which contrasts especially with the Vocational Training, GCE (General Certificate Education) and University stages, which show higher levels and are the only three stages with representation at level C2 (pioneer). As for primary school teachers, they are found between levels A2 and C2, with a large representation at level B1, around 47% of the sample at this stage.

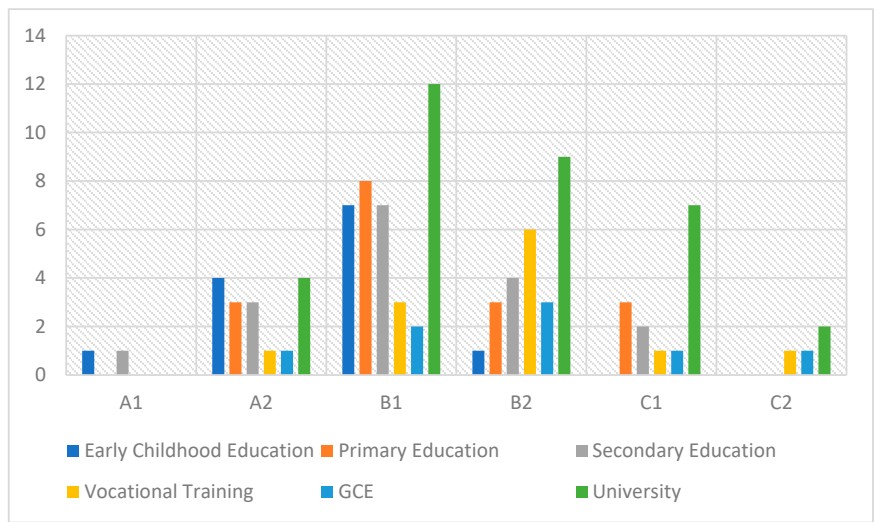

**Figure 4.** Level of Digital Competence by Stage.

Based on the results obtained by teachers working in Secondary Education, they are the only group together with Early Childhood Education teachers represented at A1 level. On the other hand, more than 60% of the sample has an intermediate level of competence (B1 and B2). Likewise, around 60% of GCE, Vocational Training, and University teachers have an intermediate level of competence (B1 and B2). Particularly relevant is the case of university teachers, where 25% of the sample obtained high scores in teaching digital competence (levels C1 and C2).

Therefore, it seems clear that teachers at higher levels of education show, in general, higher levels of digital competence in teaching than those at lower levels in the sample.

### 3.4. Results by Teaching Experience

In terms of teaching experience, as shown in Figure 5, those between 1 and 5 years of teaching experience are distributed across all levels of competence, with a particular concentration at intermediate levels; it is also one of the only groups with a presence at A1 and C2 levels. Those between 6 and 10 years of experience are represented at levels A2, B1, B2, and C1, with no representation at either the entry level or the highest proficiency level. Teachers in the sample with between 11 and 15 years of teaching experience are mainly represented at levels A2 and B1, with a low presence at level B2 and level C1; on the other hand, they are not represented at A1 or C2 levels.

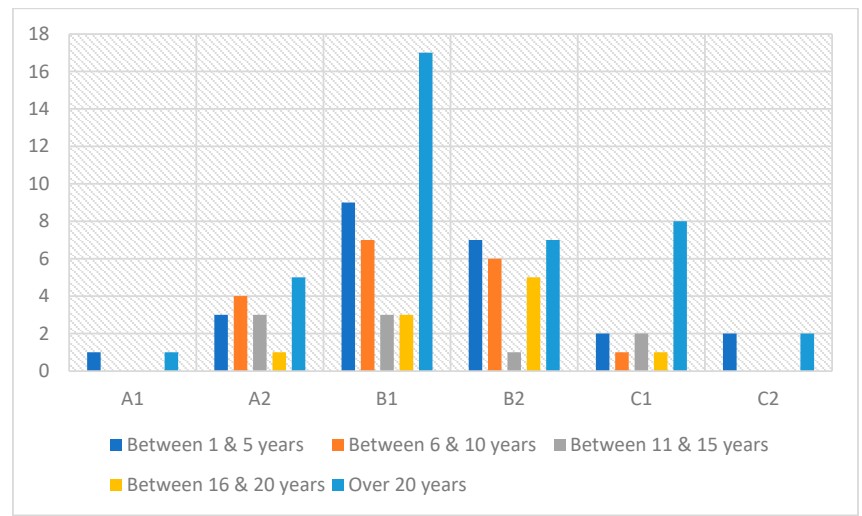

**Figure 5.** Level of Digital Competence by Teaching Experience.

The sample of teachers with between 16 and 20 years of teaching experience are highly represented at intermediate (B1 and B2); however, no data appear at A1 and C2 levels, and the sample at A2 and C1 levels is minimal. Finally, teachers with more than 20 years of teaching experience are concentrated in the intermediate and higher levels, the other group with polarised representation at A1 and C2 levels.

To sum up, we can say that the majority of the sample is concentrated in the intermediate and upper-intermediate or fist higher level (B1, B2, and C1), with hardly any representation at the lowest and highest levels.

### 3.5. Results by Type of Educational Establishment

If we analyse the type of educational establishment where the teachers carry out their professional activity (Figure 6), we found that those who work in private-state subsidised establishments show scores at the B2 level at the most, while teachers who carry out their teaching activity in public entities have higher competence levels and are the only ones who score at the higher levels (C1 and C2), although more than 60% are at the B1 and B2 stages, corresponding to those who are called integrators and experts.

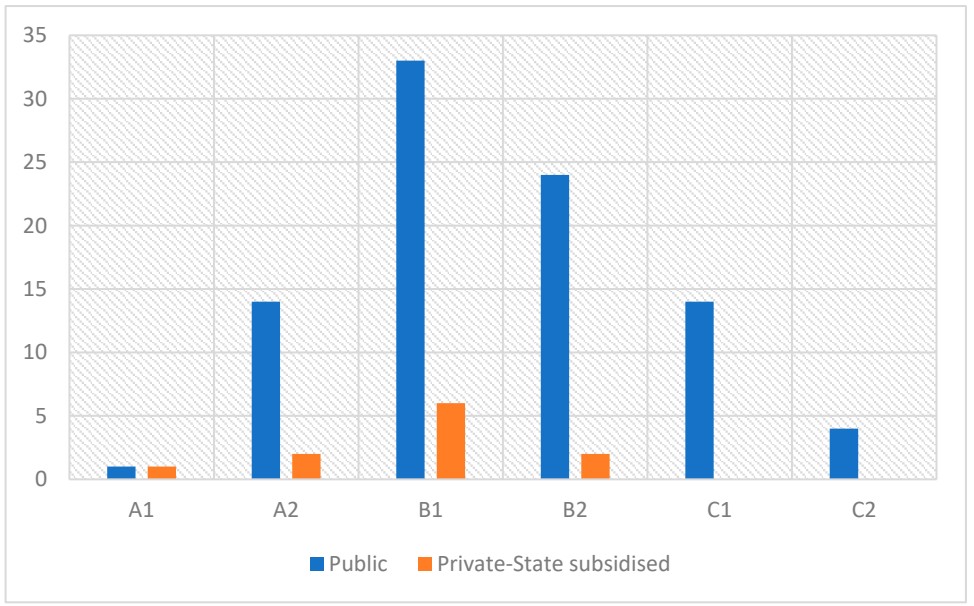

**Figure 6.** Level of Digital Competence by type of Educational Establishment.

The sample is even at A1 level, with few teachers represented at this stage. We also note the absence of teachers from private-state subsidised schools at level C, where data were collected on teachers from public schools.

### 3.6. Emerging Technologies and Their Link to Digital Competence in Teaching

As can be seen in Table 3, those participants with a low level of digital competence (A1 and A2) have not used emerging technologies at any time during their teaching. As for teachers with an intermediate level (B1 and B2), most of them do not use emerging technologies to carry out their practice, although the number of participants who do use this type of technology in the classroom increases notably at the highest level (B2). Finally, teachers with a C1 level in Digital Competence in Teaching use emerging technologies, with few exceptions, something that is evident among those with the highest competence level C2, who have all used emerging technologies at some point in time to favour the teaching–learning process of their students.

**Table 3.** Relationship between levels of digital competence and the use of emerging technologies.

| Use of Emerging Technologies | Level A | | | | Level B | | | | Level C | | | | Total | |
| | A1 | | A2 | | B1 | | B2 | | C1 | | C2 | | | |
| | N | % | N | % | N | % | N | % | N | % | N | % | N | % |
| They have used emerging technologies | 0 | 0% | 0 | 0% | 5 | 4.9% | 16 | 15.8% | 13 | 12.9% | 4 | 4% | 38 | 37.6% |
| They have not used emerging technologies | 2 | 2% | 16 | 15.8% | 34 | 33.7% | 10 | 9.9% | 1 | 1% | 0 | 0% | 63 | 62.4% |
| Total | 2 | 100% | 16 | 100% | 39 | 100% | 26 | 100% | 14 | 100% | 4 | 100% | 101 | 100% |

The type of technology most commonly used has been analysed depending on whether emerging technologies have been used in teaching. Table 4 shows that Augmented Reality and Virtual Reality are the most widely used among those who use these tools for teaching, with over 80% of teachers using them in both cases. On the other hand, Mixed Reality is the least used, with only 40% of people having at some point used this type of emerging technology to carry out activities in their classroom.

**Table 4.** Type of Emerging Technology by use.

| Use of Emerging Technologies | Augmented Reality | | | | Virtual Reality | | | | Mixed Reality | | | | Total | |
| | Control: | | No | | Control: | | No | | Control: | | No | | | |
| | N | % | N | % | N | % | N | % | N | % | N | % | N | % |
| They have used emerging technologies | 31 | 81.6% | 7 | 18.4% | 32 | 84.2% | 6 | 5.9% | 15 | 39.5% | 23 | 60.5% | 38 | 37.6% |
| They have not used emerging technologies | 0 | 0% | 63 | 100% | 0 | 0% | 63 | 100% | 0 | 0% | 63 | 100% | 63 | 62.4% |
| Total | | | | | | | | | | | | | 101 | 100% |

In Figure 7, we can see the skill level of the participants in the sample with respect to emerging technologies according to whether they have used them in their teaching activity; if they have not used them, their skill level is indicated as null.

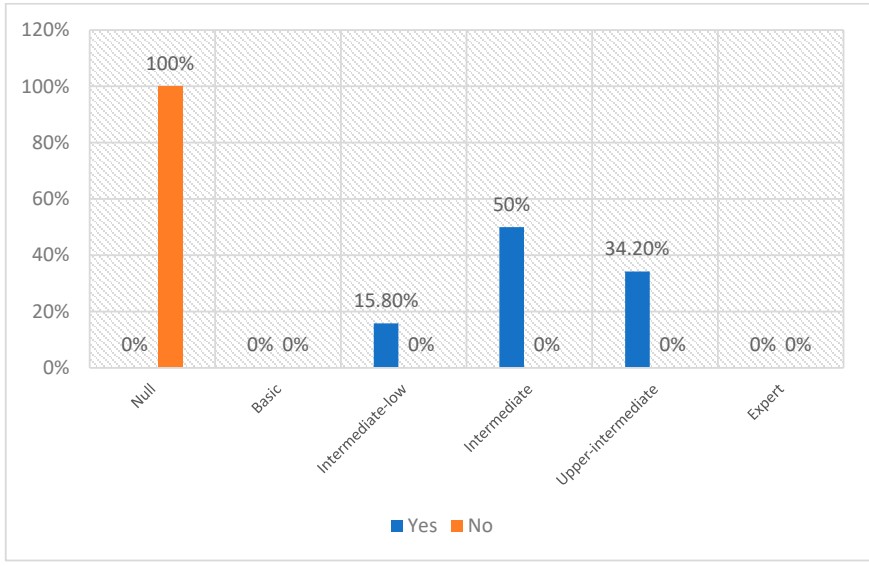

**Figure 7.** Skill level by use of Emerging Technologies.

In the case of those who have used emerging technologies, in any of their varieties (Augmented Reality, Virtual Reality, or Mixed Reality), they are classified according to their own level of perceived mastery. As can be seen, more than half of the users of this technology have an intermediate level, while nearly 34% consider their level of skill with these tools to be intermediate-high, and only 15.8% consider it to be intermediate-low. It is remarkable that none of the participants consider themselves to be experts in the handling of these emerging technologies.

## 4. Discussion and Conclusions

This study provides relevant information about the degree of digital competence among teachers at different educational stages, types of school, and countries. It also attempts to provide a broader view by showing the differences that exist according to gender, age, teaching experience, the type of establishment in which they work, or the educational stage at which they carry out their work. Similarly, it includes the variable of emerging technologies, linking their use to the levels of competence shown by the teachers in the sample, in order to provide an overall picture of teachers' skills and the impact of this type of technology on their daily activity.

The topic addressed in the article is related to the current educational reality; ICT and emerging technologies have emerged as one of the main tools for education and, therefore, digital competence in teaching has become one of the most pressing needs in the training and development of teaching activity. Moreover, one of the pillars on which the results presented above are based, and which did not appear in previous research, is the relationship between digital competence in teaching and the inclusion and use of emerging technologies in educational action. Likewise, and with the aim of proposing different guidelines to improve and continue research in the future, it would be advisable to carry out tests to establish an objective assessment of teachers' digital competences, and to compare the data observed with those perceived by teachers. It might also be interesting to carry out a face-to-face interview with the teachers so that they could qualify or complement the answers and even ask them other types of questions that would provide more in-depth knowledge of their handling and use of the technologies. After analysing the data and establishing levels of Digital Competence in Teaching through the use of the chosen tool [7], the good level of competence shown by the teachers participating in the study is evident, with a medium level of competence (integrator and expert) and a small part of the sample with a pioneer level (C2), doubling the results of those who are in the A1 stage (beginner). These data are confirmed by the similarities in the level of competence with other studies [11], in which teachers presented average levels (B1 and B2), and there were few participants who presented an A1 or C2 level.

Likewise, referring to previous studies [11], we observe that, in general terms, the sample shows higher averages in the different competence areas compared to the reference study. Therefore, in general terms, the observed sample shows similarities with the baseline studies [7,8,11], although it has slightly higher averages in all the observed competence areas. In line with previous research [7] and referring to the fields of competence described in the questionnaire, the sample shows slightly higher mean values than those obtained by other researchers, although these were limited to a specific stage, vocational training. However, the fields of competence in which teachers show the greatest weaknesses are assessment and feedback, and facilitating students' digital competence.

On the other hand, and referring to previous research [8], it is women who show a higher overall competence level, to the detriment of men; this may also be due to the existing unequal gender distribution, with female teachers making up the bulk of the sample, as well as being the most represented in the schools.

As previously stated [1], digital competence is one of the main workhorses of both teachers and educational bodies and the laws that govern education systems, since it is never possible to achieve the complete attainment of the proposed objectives, although progress remains constant and significant.

Emerging technologies are here to stay and substantially modify both learning spaces and methodologies, and although there is still a long way to go, given that their use is not yet widespread, it is important to continue promoting continuous training in this field for both teachers and students. We must not forget that the application of technology depends not only on its availability, but also on the ability to maximise the performance and possibilities of technology by training teachers and facilitating its use in the classroom. In the same way and as previous studies have shown [15,16], teachers are obviously not adapted to the current technological reality, and do not effectively integrate emerging technologies into the teaching–learning processes despite the fact that these technologies are increasingly present in society. Moreover, those who do still need to improve their skills with these technologies in order to maximise the opportunities they provide for the improvement of teaching practice.

It is also important that scientific literature continues to be produced on the subject in order to create knowledge and promote the improvement of the use of these tools, the processes necessary for the acquisition of the necessary skills for their use, and the development of new methodologies that integrate emerging technologies. On the other hand, and based on the results obtained in the questionnaire, it seems clear that there is an urgent need to improve averages in the fields of assessment and feedback, as well as in facilitating students' digital competence, both of which can be improved and encouraged by providing training and alternatives for teachers. Especially the field of competence related to students, this is one of the great deficits observed in the study which urgently needs to be solved, as it is important to provide correct training for students in the digital sphere in order to favour and optimise their incorporation into the digitalised world, the reality in which we live and in which they will have to carry out their professional work.

In conclusion, and referring to previous studies [15], the training of future teachers is also one of the main instruments that educational institutions should use to improve and adapt teaching practice to the technological reality of the 21st century and the large number of tools provided by the many technological advances.

To sum up, and as the authors show in their study [17], there are still milestones to be reached in terms of digital competence in teaching, although it is clear that there has been a strong development of teachers' competences, with great progress being made, and that the results will soon become evident as a response to the new challenges posed.

**Author Contributions:** Conceptualisation: M.Á.G.-D., S.R.-C., V.D.-B. and M.L.-Á.; Investigation: M.Á.G.-D., S.R.-C., V.D.-B. and M.L.-Á.; Resources: M.Á.G.-D., S.R.-C., V.D.-B. and M.L.-Á.; Writing—original draft preparation: M.Á.G.-D.; Writing—review and editing: M.Á.G.-D., S.R.-C., V.D.-B. and M.L.-Á. All authors have read and agreed to the published version of the manuscript.

**Funding:** This research received no external funding. It is part of the Doctoral Thesis: "La competencia digital docente y la creatividad en docentes de distintos niveles educativos" (Digital competence in teaching and creativity in teachers at various levels of education).

**Institutional Review Board Statement:** The study was conducted according to the guidelines of the Declaration of Helsinki, and approved by the Ethics Committee of University of Burgos (IR 3/2023; date of approval: 7 February 2023).

**Informed Consent Statement:** Informed consent was obtained from all subjects involved in the study.

**Data Availability Statement:** Due to privacy and confidentiality issues the data are not available.

**Acknowledgments:** 

**Conflicts of Interest:** The authors declare no conflict of interest.

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
