# Peer review of "Emerging Technologies and Their Link to Digital Competence in Teaching"

_futureinternet, doi:10.3390/fi15040140_

Round 1

Reviewer 1 Report

The main question addressed by the research is the assessment of the level of digital competence of teachers in Spain, Portugal, and Chile, using the "DigCompEdu Check-In" questionnaire and their use of emerging technologies in the teaching-learning process. The results of the evaluation are convincing. The study highlights the urgent need to improve facilitating students' digital competence.

However, the article does not discuss the originality of the topic or how it adds to the subject area compared to previous research. Related studies can be highlighted in the introduction section.

Regards

Author Response

First of all, we would like to thank you for your assessment and for taking the time to make your comments, as they allow us to substantially improve the content and understanding of the article. Likewise, we thank you for your opinion on the article, and for your comment on the urgent need to promote digital competence in active teachers and in teacher training, and also the need to include emerging technologies in training in order to provide more opportunities in the future for teachers and, therefore, also for students and improve their training.

On the other hand, in response to the proposed improvements, the assessment of the originality and importance of the topic addressed and the possible contributions in comparison with previous studies has been included in the discussion and conclusions. Also, some of the most relevant studies have been highlighted in the introduction to demonstrate their importance.

Reviewer 2 Report

The paper deals with the interesting topic of the use of emerging technologies by teachers in the utilization process at various levels within schools. The authors present the results based on the DigCompEdu questionnaire, which provides a subjective self-assessment of teachers about the quality of using digital competences and using tools in teaching. For the article, it would be appropriate to compare the participants' actual digital competences obtained on the basis of objective tests and compare them with self-assessment. In the article, it would be appropriate to state what are the possibilities of objective assessment of digital skills, because in most cases self-assessment is slightly overestimated.

On the other hand, the results of the questionnaires are well evaluated in the article.

As part of the evaluation, it would be appropriate not to draw conclusions based on the results in section 3.5 from line 274, since it is not possible to compare the same or approximately the same size sample of participants from the public sector and private-state subsidized.

Please add its meaning in brackets on line 233 after the abbreviation GCE.

Author Response

First of all, we would like to thank you for your interesting opinions and evaluations of our article, as well as for the time invested in making your observations, which allow us to improve and enhance the understanding of the work carried out.

In response to the observations made, we have included as a proposal for improvement and future line of research the possibility of comparing self-perception with the performance of objective tests to assess the use of emerging technologies and digital skills of teachers, on the other hand, we have mentioned the possible tests to objectively know these data and provide a starting point for further research.

With regard to not drawing conclusions about the comparison of teachers in public and private schools, it was simply an attempt to describe the sample, and it became clear that it was impossible to make a comparison due to the disparity of the sample obtained. We have also added the meaning of the abbreviation you mentioned.

Reviewer 3 Report

In this manuscript, the authors assess the competence level of teachers using a descriptive study. The results are supported by test results. The impact of emerging technologies is confirmed in terms of teachers’ skills. Overall, I recommend it for publication after major revisions:
1.    In the introduction part, the authors mention that the emerging technologies include augmented reality, virtual reality, and mixed reality. Although this manuscript is based on a lot of questionnaires, I suggest adding some technical papers related with AR/VR to supplement technical backgrounds.
2.    In the 356th line, the authors explain that the unequal gender distribution leads to the result that women show a higher overall competence level. Is it possible to make the number of women and men equal? In this way the research result may be convincing. Otherwise, the result would be barely supported by the questionnaire.
3.    In this paper, the title is called “emerging technologies and their link to digital competence in teaching”. However, there is just a section 3.5 that focuses on the content of the title. From my perspective, more contents regarding the emerging technologies should be added. For example, do some teachers really understand the difference between AR/VR and MR? In your questionnaire, how do you explain these technologies that may be unfamiliar to some teachers?
4.    In your opinion, how do you further improve the design for your questionnaires? Some factors and analysis are too straightforward. Some of the conclusions may be known before the questionnaires are returned.

Author Response

First of all, we would like to thank you for your work reviewing our article and we are grateful for all the comments you have made, as we believe that they substantially improve the content and enhance the understanding of the work presented.

In order to respond to the proposed corrections, some technical articles on VR, AR and MR have been included to allow the reader to obtain a globalised idea of the emerging technologies. They have been included in line number 47 till 58.  

We thank you for your contribution regarding the unequal gender distribution, but by using the validated DigCompEdu questionnaire, a descriptive analysis of the sample has been carried out, not to establish correlations between the different sexes, but simply to describe the reality of the sample we have. Clearly, an attempt could be made to balance the existing disparity by carrying out a purposive sampling and choosing the sample, although this process could condition the results with respect to digital competence and the use of emerging technologies.

Likewise, in order to try to alleviate one of the weak points it proposes, an attempt has been made to increase the weight of the contents related to emerging technologies that allow the reader to obtain more relevant data on the subject in order to favour the understanding of the results presented in the article. Likewise, the questionnaire mentioned the difference between AR, VR and MR, so that all the participating teachers could understand the content of the question posed.

Round 2

Reviewer 3 Report

Thank you for your revision. Now It is acceptable.